# Digestive glycosidases from cannonball jellyfish (*Stomolophus* sp. 2): identification and temporal-spatial variability

Raul Balam Martinez-Perez[1,2,*], Jorge A. Rodriguez[2,*], Miguel A. Cisneros-Mata[3], Luis Alonso Leyva Soto[1,4], Pablo Gortáres-Moroyoqui[1], Ana Renteria-Mexia[1], Edna Abigail Hernandez Corral[1] and Lourdes M. Diaz-Tenorio[1]

[1] Department of Biotechnology and Food Sciences, Instituto Tecnológico de Sonora, Ciudad Obregón, Sonora, Mexico
[2] Industrial Biotechnology, Centro de Investigación y Asistencia en Tecnología y Diseño del Estado de Jalisco, A.C., Zapopan, Jalisco, Mexico
[3] Regional Center for Aquaculture and Fisheries Research, Instituto Nacional de Pesca y Acuacultura, Guaymas, Sonora, Mexico
[4] Assistance Management of Scientific Develpment, Consejo Nacional de Ciencia y Tecnología, Mexico City, Mexico City, Mexico
[*] These authors contributed equally to this work.

Corresponding author
Lourdes M. Diaz-Tenorio, lourdes.diaz@itson.edu.mx

PeerJ Hubs
Published on behalf of
International Association for Biological Oceanography
IABO

## ABSTRACT

Jellyfish are economically important organisms in diverse countries, carnivorous organisms that consume various prey (crustaceans, mollusks, bivalves, *etc.*) and dissolved carbohydrates in marine waters. This study was focused on detecting and quantifying the activity of digestive glycosidases from the cannonball jellyfish (*Stomolophus* sp. 2) to understand carbohydrate digestion and its temporal-spatial variation. Twenty-three jellyfish gastric pouches were collected in 2015 and 2016 in the Gulf of California in three localities (Las Guásimas, Hermosillo, and Caborca). Nine samples were in intra-localities from Las Guásimas. Chitinase (Ch), $\beta$-glucosidase ($\beta$-glu), and $\beta$-$N$-acetylhexosaminidase ($\beta$-NAHA) were detected in the gastric pouches. However, cellulase, exoglucanase, $\alpha$-amylase, polygalacturonase, xylanase, and $\kappa$-carrageenase were undetected. Detected enzymes showed halotolerant glycolytic activity ($i = 0$–4 M NaCl), optimal pH, and temperature at 5.0 and 30–50 °C, respectively. At least five $\beta$-glucosidase and two $\beta$-N-acetylhexosaminidase were detected using zymograms; however, the number of proteins with chitinase activity is not precise. The annual variation of cannonball jellyfish digestive glycosidases from Las Guásimas between 2015–2016 does not show significant differences despite the difference in phytoplankton measured as chlorophyll $\alpha$ (1.9 and 3.4 mg/m$^3$, respectively). In the inter-localities, the glycosidase activity was statistically different in all localities, except for $\beta$-$N$-acetylhexosaminidase activity between Caborca and Hermosillo (3,009.08 $\pm$ 87.95 and 3,101.81 $\pm$ 281.11 mU/g of the gastric pouch, respectively), with chlorophyll $\alpha$ concentrations of 2.6, 3.4 mg/m$^3$, respectively. For intra-localities, the glycosidase activity did not show significant differences, with a mean chlorophyll $\alpha$ of 1.3 $\pm$ 0.1 mg/m$^3$. These results suggest that digestive glycosidases from *Stomolophus* sp. 2 can hydrolyze several carbohydrates that may belong to their prey or carbohydrates

dissolved in marine waters, with salinity over ≥ 0.6 M NaCl and diverse temperature (4–80 °C) conditions. Also, chlorophyll $\alpha$ is related to glycosidase activity in both seasons and inter-localities, except for chitinase activity in an intra-locality (Las Guásimas).

## INTRODUCTION

The biological aspects of jellyfish are essential due to their economic and ecological worldwide importance. In Mexico, only one jellyfish genus (*Stomolophus*) is caught, processed, and commercialized. The *Stomolophus* sp. 2, commonly known as cannonball jellyfish, is widely distributed in the northwest of Mexican waters, in the Gulf of California (*Gómez-Daglio & Dawson, 2017*). From all *Stomolophus* jellyfish species, *Stomolophus* sp. 2 is easily identifiable by its intense blue color and mushroom shape (*López-Martínez & Álvarez Tello, 2013*). Given the importance of *Stomolophus* jellyfish, different efforts to know their life cycle, reproduction (*López-Martínez et al., 2023*), fecundity (*López-Martínez et al., 2023*; *Treible & Condon, 2019*), diet prey selection (*Álvarez Tello, López-Martínez & Lluch-Cota, 2016*; *Larson, 1991*), pigmentation (*Martínez-Pérez et al., 2020*), nutrition (*Camacho-Pacheco et al., 2022*), and bloom events (*Gómez-Salinas, López-Martínez & Morandini, 2021*) have been studied. However, there are few studies on the hydrolytic enzymes involved in the physiological digestive process in the genus *Stomolophus* (*Bodansky & Rose, 1922*; *González-Valdovinos, Ocampo & Tovar-Ramírez, 2019*; *Martínez-Pérez et al., 2020*). Therefore, understanding the digestion process of biomolecules by jellyfish can help to understand biological aspects such as their growth and the success of massive blooms, which can generate economic damage to many fisheries worldwide.

Hydrolases such as peptidases, lipases/esterases, and glycosidases are essential in digestion. Glycosidases (EC 3.2.1.x) are present in gastric juices from diverse marine invertebrates such as mollusks and crustaceans, hydrolyzing simple and complex carbohydrates (*Hylleberg Kristensen, 1972*). Cnidarians are considered carnivores, so limited information is known about their digestive glycosyl hydrolases. Despite this, some reports about chitinase activity in the hydras *Hydra attenuata*, *H. circuncicta*, and *Podocoryne carnea* (*Klug et al., 1984*), and chitobiose activity in gastric extracts from the sea anemone *Adamasia sulcate*, *Anthopleura balli*, and *Edwarsia callimorpha* (*Jeuniaux, 1962*) have been reported. The digestive activity of $\beta$-$N$-acetylglucosaminidase, $\beta$-$N$-acetylgalactosaminidase, $\beta$-glucosidase, and $\alpha$-manosidase was present in *Anthopleura* spp., *Cnidopus japonica*, and *Metridium senile*. However, no other enzymes were reported, such as galactosidase, xylosidase, and mannosidase (*Molodtsov & Vafina, 1972*). In jellyfish, the chitinase activity in mesoglea (*Nagai, Watarai & Suzuki, 2001*) and chitinase zymogens in the gastric cirri of *A. aurita* have been reported (*Steinmetz, 2019*). Understanding the type and kind of enzymes that jellyfish use to digest their food can help to generate new

diets that can be used to keep them in captivity so that they can be studied in controlled conditions and used for technological processes, taking advantage of the biomolecules that compose them.

It is well known that jellyfish of the genus *Stomolophus* feed on crustaceans, gastropods and mainly on bivalves, copepods, barnacles, and fish eggs (*Álvarez Tello, López-Martínez & Lluch-Cota, 2016*; *Larson, 1991*; *Padilla-Serrato et al., 2013*). However, jellyfish also ingests dissolved organic particles in water derived from the activity of marine microorganisms by the transport of organic matter from rain or groundwater (*Carlson & Hansell, 2015*), which are rich in acetates, simple sugars, and high molecular weight carbohydrates (*Mühlenbruch et al., 2018*; *Repeta et al., 2002*). Dissolved polysaccharides, as organic matter in surface waters, range from 0.3 to 210 µM of carbon and show seasonal variation (*Gui-Peng et al., 2010*), and have been monitored as chlorophyll $\alpha$ in water (*Sánchez-Pérez et al., 2020*).

Regarding juvenile and adult forms, most jellyfish species have existed virtually without change for over 600 million years and are active, voracious, opportunistic predators (*Hay, 2006*). Polyps are also active feeders and plankton predators able to store energy to survive asexually reproducing for years (*Van Walraven, Van Bleijswijk & Van der Veer, 2020*; *Pengpeng et al., 2021*). Hence, given sufficient food items, it is unlikely that the type of biota would contribute to an adaptation process. Adaptation may result from jellyfish's inherent capability to survive under extreme conditions such as global warming and other anthropic effects such as pollution (*Lee et al., 2023*).

To date, no studies on how the jellyfish spatial–temporal events may modify digestive glycolytic activity patterns, so the present study was focused on evaluating the temporal-spatial regulation of digestive enzymes in cannonball jellyfish (*Stomolophus* sp. 2) and its biochemical characteristics in the gulf of California, to provide an overview of the adult jellyfish nutritional needs, feed, digestive physiology, and massive blooms compression, concerning the carbohydrates that they digest.

## MATERIAL AND METHODS

### Sample collection

Twenty-three specimens of *Stomolophus* sp. 2 were collected on the coast of Las Guásimas (27°41′59.4″N, 110°36′30.2″W) in April 2015 and were used to obtain operational parameters and individual variability of glycosidases in jellyfish gastric pouches. For the annual evaluation of glycosidases in gastric pouches, 23 specimens from Las Guásimas were collected (27°46′50.9″N, 110°37′3.4″W) in February 2016, which were used for the evaluation of inter-locality. Likewise, 23 specimens were collected from the Hermosillo coast (28°28′12.9″N, 111°43′7.86″W) in April 2016 and the coast of Caborca (30°55′26.34″N, 113°05′7.98″W) in May 2016. For the evaluation of intra-locality, three other specimens were collected from three geographical points (27°46′17.46″N, 110°36′59.4″W; 27°45′37.32″N, 110°36′45.84″; 27°43′59.4″N, 110°36′44.46″W) from Las Guásimas, in April 2016 (Fig. 1A).

The jellyfish sampled from Las Guásimas in 2015 measured each organism's bell diameter and mass immediately after the catch. A cross-shaped cut in the bell was carried out to

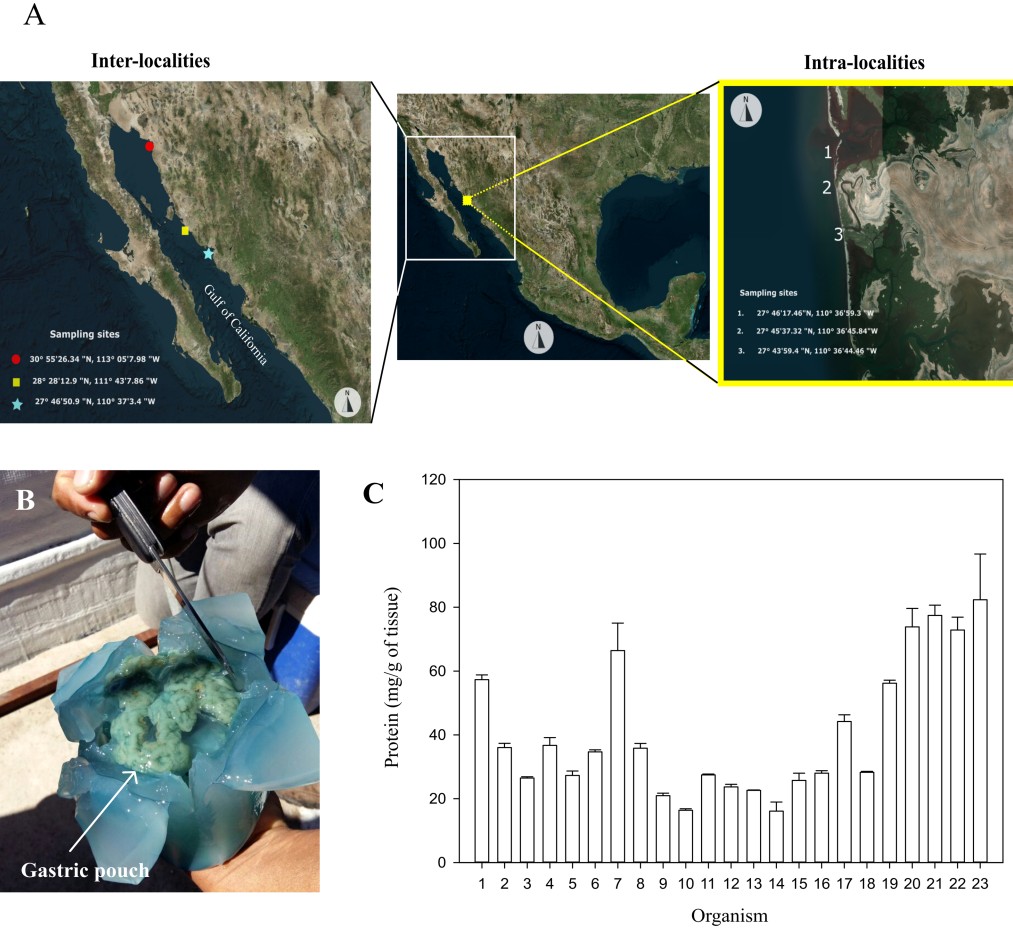

**Figure 1** **Sampling sites, dissection, and protein of gastric pouches from _Stomolophus_ sp. 2.** (A) Sample collection. (B) X-shaped cut in the jellyfish bell to expose gastric pouches. (C) Quantification of soluble protein in the gastric pouches of each of the 23 from Las Guásimas in 2015. The white arrow indicates the position of the gastric pouches; each sample was quantified in triplicate (mean ± S.D.).

obtain gastric pouches. A fold was made in each cut to expose the gastric pouches (Gp) and then removed manually. The Gp was immediately stored at 0 °C, transported to the laboratory, stored at −20 °C, and lyophilized for further analysis.

## Enzymatic extracts from gastric pouches and protein quantification

Each lyophilized sample of the jellyfish gastric pouch (30–50 mg) was resuspended in one mL of distilled water for 30 min at 0 °C with low vortexing every 5 min. After incubation, the samples were centrifuged at 9,000 $g$, 4 °C, for 10 min. The supernatant was recovered and stored at −20 °C for further analysis. The above procedure was performed in the same way to prepare an enzyme extract from a mixture of the 23 jellyfish gastric pouches samples (50 mg of tissue from each organism). Protein was quantified using the Bradford method using bovine serum albumin (BSA) (1 mg/mL) as a standard protein (_Bradford, 1976_).

### Enzymatic activity evaluation
### Glycosidase activity with natural substrates

The activity of amylase, pectinase, chitinase, carrageenase, xylanase, exo-, and endoglucanase was quantified by measuring reducing sugars released by enzymatic hydrolysis. A colorimetric method was used, using 3,5-dinitrosalicylic acid (DNS) (*Miller, 1959*), and the substrates used were 0.5% soluble starch (amylase activity), 0.5% apple pectin (pectinase activity), 1.0% carboxymethyl cellulose (CMC activity; endoglucanase activity), 1.0% microcrystalline cellulose (exoglucanase activity), 1.0% colloidal chitin (chitinase activity), 0.5% carrageenan (carrageenase activity) and 0.5% xylan (xylanase activity) in a w/v ratio. Briefly, 50 µL of the enzymatic extract was mixed with the corresponding buffer in a 1:1 v/v ratio and 500 µL substrate at 37 °C. Then, each sample with the substrate was mixed at 100 rpm by time incubation intervals (1, 5, and 24 h). Aliquots of 50 µL were taken for absorbance measurements at 540 nm in a microplate reader Molecular Devices SpectraMax 190.

### $\beta$-N-acetylhexosaminidase activity

To quantify $\beta$-$N$-Acetylhexosaminidase activity, the enzymatic extract (40 µL) was mixed with 40 µL of phosphate buffer (0.2 M, pH 6.0) and 40 µL of substrate $p$-nitrophenyl-$\beta$-$N$-acetylglucosaminide (1 mg/mL). The reaction mixture was incubated at 37 °C for 15 min (*Tronsmo & Harman, 1993*), then 0.5 mL of 0.02 M NaOH was added, and the absorbance was recorded at 405 nm in a microplate reader Molecular Devices SpectraMax 190, adding 200 uL of the sample. A standard curve of p-nitrophenol (0–100 µM) was performed.

### $\beta$-glucosidase activity

The activity was quantified using $p$-nitrophenol $\beta$-$D$-glucopyranoside as a substrate. The reaction mixture was 270 µL of the substrate (1 mg/mL) and 30 µL of enzyme extract, incubated at 37 °C and shaken at 100 rpm for 60 min, then 300 µL of 2% $Na_2CO_3$ was added, and the absorbance was recorded at 410 nm (*Takashima et al., 1999*), then, 200 µL of the sample were used to record the absorbance in a microplate reader Molecular Devices Spectramax 190. A standard curve of $p$-nitrophenol was made from 0 to 100 µM.

### Effect of temperature, pH, and ionic strength

The effect of temperature on the enzyme activity was evaluated by incubating the reaction mixture (substrate and enzyme) at different temperatures (4, 25, 40, 50, 60, 70, and 80 °C) for 1 h; then, the absorbance was measured to quantify the enzyme activity. The effect of pH was evaluated by incubating the reaction mixture between 5.0 and 10.0 for 1 h, then enzymatic activity was measured in a microplate reader Molecular Devices Spectramax 190. Finally, the ionic strength effect was evaluated between 1.0 and 4.0 M NaCl. The buffers used were 100 mM citrate in the range between pH 3.0 to 5.0, 0.2 M phosphate buffer at pH 6.0, 50 mM Tris–HCl buffer for the range between pH 7.0 and 9.0, and 0.2 M carbonate-bicarbonate buffer for pH 10.0 were used.

## Protein SDS-PAGE analysis

To determine the protein profile of gastric pouches, proteins were separated by electrophoresis, according to *Laemmli (1970)*. A gel of 12% sodium dodecyl sulfate-polyacrylamide (SDS-PAGE) was made to study the protein profile from gastric pouches samples. The gel was loaded with 11 µg of the enzymatic extract from a mixture of the 23 organisms in a 1:1 ratio with a loading buffer (0.125 M Tris–HCl, 4% SDS, 20% v/v glycerol, and 0.02% blue bromophenol). A low molecular weight marker (1610304; Bio-Rad, Hercules, CA, USA) was used to identify the molecular weight of the gastric pouch proteins.

Protein separation was performed at 7 mA and 4 °C in an electrophoresis chamber Hoeffer. After electrophoresis, the gel was removed and stained with a Coomassie solution (0.05% Coomassie Blue R-250, 40% methanol, and 7% acetic acid) for 1 h. Subsequently, the gel was transferred to a staining solution (the same solution without dye). The gels were visualized using a Gel Doc EZ system (Bio-Rad, Hercules, CA, USA).

## Glycosidases zymograms

Exoglucanase, $\beta$-glucosidase, and chitinase activity were evaluated on a 12% SDS-PAGE polyacrylamide gel (*Bai et al., 2013*). After protein separation, the gel was washed with distilled water and incubated in 30 mL of buffer solution (50 mM Tris–HCl, pH 6.0) for 1 h. The gel was then transferred to a buffer containing 100 mM citrate, pH 5.0, 2.5% Triton X-100 for 60 min, then incubated with 15 mL of fresh citrate buffer with 4-methylumbelliferyl-glucopyranoside (4-MUGLc) (w/v); in the case of $\beta$-glucosidase,15 mL of citrate buffer, pH 6.0 with 4-methylumbelliferil cellobioside (4-MUG) (w/v) was used for exoglucanase activity. Incubations were at 37 °C with agitation (100 rpm) for 10 min, finally visualized with UV light (302 nm). For chitinase activity, 0.01% of glycol chitin was copolymerized with the SDS-PAGE, washed according to the above mentioned conditions, then incubated in a solution containing 0.01% Calcofluor White M2R in 0.5 M Tris–HCl pH 8.0, next, incubated for 1 h with distilled water at room temperature (25 °C) and photo-documented under UV light (302 nm) (*Tronsmo & Harman, 1993*).

The zymogram was carried out for amylase activity using a 12% SDS-PAGE gel. After electrophoresis, the gel was washed with distilled water. Later, the gel was incubated in a 50 mM sodium acetate buffer at pH 5.0 for 60 min and kept at 4 °C for 12 h in a 50 mM sodium acetate buffer solution at pH 5.0 with 0.5% soluble starch. Next, the gel was incubated at 37 °C for 120 min. A Lugol stain solution was used to detect amylase activity bands (2% iodine and 4% potassium iodide in distilled water (*Wanderley et al., 2004*)).

## Average surface temperature and chlorophyll $\alpha$

The data of surface temperature (°C) and chlorophyll $\alpha$ (mg/m$^3$) were obtained from the GIOVANNI 4.21 web server of the National Aeronautics and Space Administration (NASA; https://giovanni.gsfc.nasa.gov/giovanni/). Data obtained were from the jellyfish sampling sites between 2015 and 2016. The chlorophyll $\alpha$ concentration and surface temperature data were obtained using NASA's medium resolution imaging spectroradiometer (MODIS web; https://modis.gsfc.nasa.gov/).

**Table 1  Gastric pouch glycosidase activity in *Stomolophus* sp. 2.**

| Enzyme | Specific activity (mU/g dry tissue) |
|---|---|
| Chitinase | 24.28 ± 0.38 |
| $\beta$-$N$-acetylhexosaminidase | 3645.00 ± 198.10 |
| Cellulase | N/D |
| Exoglucanase | N/D |
| $\beta$-glucosidase | 374.00 ± 10.30 |
| $\alpha$-amylase | N/D |
| Polygalacturonase | N/D |
| Carragenase | N/D |
| Xylanase | N/D |

Notes.

Quantification of enzyme activity was performed with the mixture of 23 individuals of *Stomolophus* sp. 2 (Las Guásimas, 2015).

N/D,  No activity detected.

Mean ± SD.

## Statistical analysis

Data for enzymatic activity was presented as mean ± S.D. The statistical test for medians in different years of the catch was performed using the one-way ANOVA test ($p < 0.05$). Differences between sites of jellyfish catches were calculated using the Kruskal-Wallis test ($p < 0.05$) and the Tukey's honestly significant difference (HSD). All analyses were done using Statgraphics Plus 5.1 software and analyzed in triplicate.

## RESULTS

### Glycosidase activity in gastric pouches of *Stomolophus* sp. 2

The protein content of the gastric pouches was 40.73 ±2.47 mg of protein per g of dry tissue of gastric pouches (Figs. 1B–1C). The glycosidase activity in the mixture of gastric pouches from 23 organisms of *Stomolophus* sp. 2 catches in Las Guásimas in 2015  showed chitinase (24.28 ±0.38), $\beta$-$N$-acetylhexosaminidase (3,645.00 ±198.10), and $\beta$-glucosidase (374.00 ±10.30) activities (Table 1). However, no amylase, pectinase, carrageenanase, xylanase, exoglucanase, and endoglucanase activities were detected under all conditions assayed (Table 1).

The mean of individual $\beta$-glucosidase activity was 33.76 ±3.72 mU/g of dry tissue; this enzyme was detected only in eight of the 23 organisms between 56.90–154.51 mU/g of dry tissue (Fig. 2A), showing significant differences in three of the eight organisms. In comparison, the mean of chitinase activity was 31.13 ±4.42 mU/g of dry tissue and was detected in 10 of the 23 organisms in a range between 6.85–41.82 mU/g of dry tissue (Fig. 2B), where only two of ten organisms showed significant differences. However, the mean $\beta$-$N$-acetylhexosaminidase activity was 5,303.67 ±381.04 mU/g of dry tissue. It was found in all samples in a range between 1,277.35–8,865.63 mU/g of dry tissue (Fig. 2C), showing a more significant number of differences between the enzyme activities of the organisms sampled. Significant differences between 2015 and 2016 in digestive glucosidase activity are shown in Table 2.

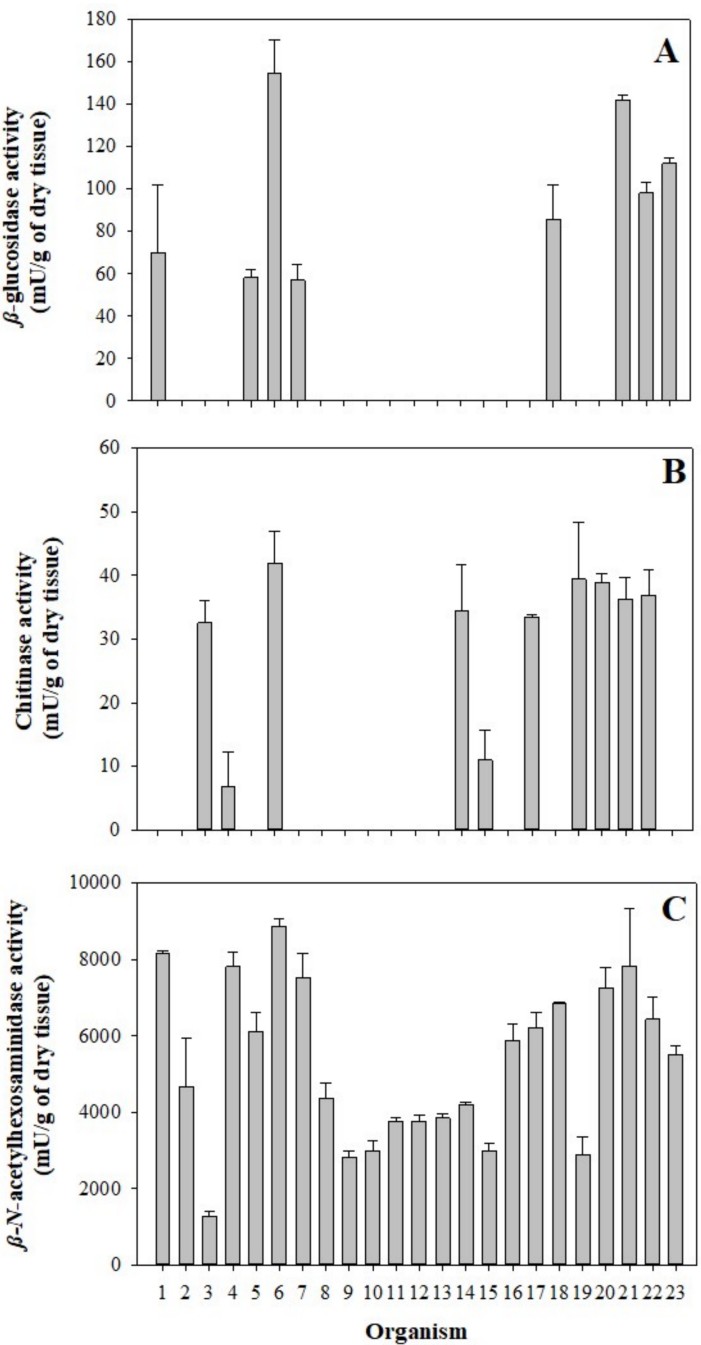

**Figure 2** **Glycolytic activity in the gastric pouch from *Stomolophus* sp. 2.** (A) $\beta$-glucosidase activity; (B) Chitinase activity, and (C) $\beta$-$N$-acetylhexosaminidase. Bars are represented by mean $\pm$ SD.

The effect of NaCl in digestive glycosidases present in gastric pouches showed halotolerant potential. Chitinase and $\beta$-$N$-acetylhexosaminidase increase at ionic strength 2.28 and 2.0-fold in 3.0 M and 1.0 M of NaCl, respectively, maintaining their activity 1.86-fold in 4.0 M NaCl for chitinase. At the same time, $\beta$-$N$-acetylhexosaminidase at 4.0

**Table 2  Annual variation of gastric sac carbohydrase activity in *Stomolophus* sp. 2.**

| Year | Enzymatic activity (mU/g of tissue) | | |
|------|-------------|------------------------------|---------------|
| | **Chitinase** | **$\beta$-$N$-acetylhexosaminidase** | **$\beta$-glucosidase** |
| 2015 | 24.28 ± 0.38[a] | 3645.00 ± 198.10[c] | 374.00 ± 10.30[d] |
| 2016 | 36.89 ± 0.90[b] | 3848.24 ± 156.56[c] | 2195.44 ± 84.20[e] |

**Notes.**

Mean ± SD, Different letters in each column indicate statistical differences.

**Table 3  Annual variation factors of glycosidase activity.**

| Year | Diameter of bell[*] (cm) | Superficial water temperatura (°C) | Chlorophyll $\alpha$ (mg/m$^3$) |
|------|----------------|------------------|-----------------|
| 2015 | 8.8 ± 2.6[a] | 23.6 | 1.9 |
| 2016 | 12.2 ± 0.8[b] | 20.6 | 3.4 |

**Notes.**

*Different letters in the column indicate statistically different years. Chlorophyll $\alpha$ data obtained from GIOVANNI 4.21.

M NaCl decreased its activity by up to 15%. For $\beta$-glucosidase, the activity increases as ionic strength increases, reaching a 1.46-fold activity at 4.0 M NaCl (Fig. 3A).

Results of the pH effect showed that glycosidases work at acidic pH (5.0); however, chitinase works better in alkaline conditions (8.0–10.0), while $\beta$-$N$-acetylhexosaminidase and $\beta$-glucosidase decreased their activity above pH 6.0 until losing 93.75 ±0.27% and 92.82 ±1.17% of their activity (Fig. 3B). Temperature plays an essential role in the activity of digestive enzymes. In *Stomolophus* sp. 2, glycosidases show different optimum temperatures, for $\beta$-$N$-acetylhexosaminidase was 40 °C, while 50 °C and 30 °C for $\beta$-glucosidase and chitinase, respectively. Above these temperatures, the activity decreased to 0.0 ±0.0%, 50.04 ±2.22%, and 22.36 ±1.13% for chitinase, $\beta$-$N$-acetylhexosaminidase and $\beta$-glucosidase, respectively (Fig. 3C).

The identification of the protein profile and glycosidases in gastric pouches was monitored by SDS-PAGE (12%). Several proteins were present in all gastric pouches, and a band of 30 kDa in almost all organisms was detected (Fig. 4A). In Fig. 4B, $\beta$-$N$-acetylhexosaminidase activity was observed in 12 organisms (at least three bands with activity were present with a molecular mass above 45 kDa). All organisms showed activity of digestive $\beta$-glucosidase, and at least five bands were identified between 21–66 kDa (Fig. 4C). The bands corresponding to digestive chitinases are shown in Fig. 4D.

The glycosidase activity of samples from Las Guásimas (2015 and 2016) shows significant differences in the enzyme activities evaluated (Table 2); also, there is a difference between the size of the adult jellyfish sampled. The mean of the bell diameter of *Stomolophus* sp. 2 was statistically different ($p < 0.05$), 3.4 cm higher in 2016 than in 2015 (Table 3). There is also a difference in surface temperature (3 °C higher in 2015) and chlorophyll $\alpha$ concentration (1.5 mg/m$^3$ higher in 2016) on the same sample site (Table 3).

Variations in glycosidase activity in the three inter-local geographical locations (Las Guásimas, Hermosillo, and Caborca in 2016) were evaluated. Glycosidase activity was statistically different ($p < 0.05$) in organisms from inter-localities (Las Guásimas, Hermosillo, and Caborca) (Table 4). The mean of the bell diameter was statistically

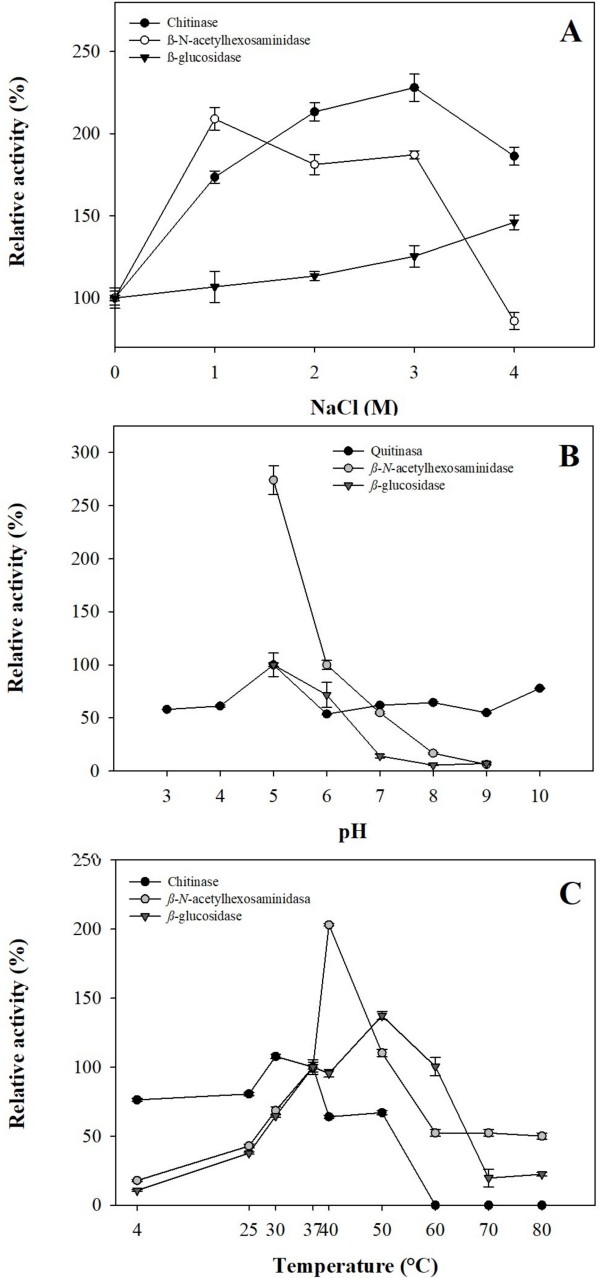

**Figure 3** **Operational parameters from glycosidases from gastric pouches of *Stomolophus* sp. 2.** (A) Ionic strength with NaCl; (B) optimum pH, and (C) optimum temperature. Each analysis was performed in triplicate (mean ± SD).

different in the three localities, being notably higher in Las Guásimas (12.2 cm) than in Hermosillo (7.9 cm) and Caborca (9.1 cm), while the chlorophyll $\alpha$ concentration values were the same in Las Guásimas and Caborca (3.4 mg/m$^3$), being higher than the concentration in Hermosillo (2.6 mg/m$^3$). On the other hand, the surface temperature
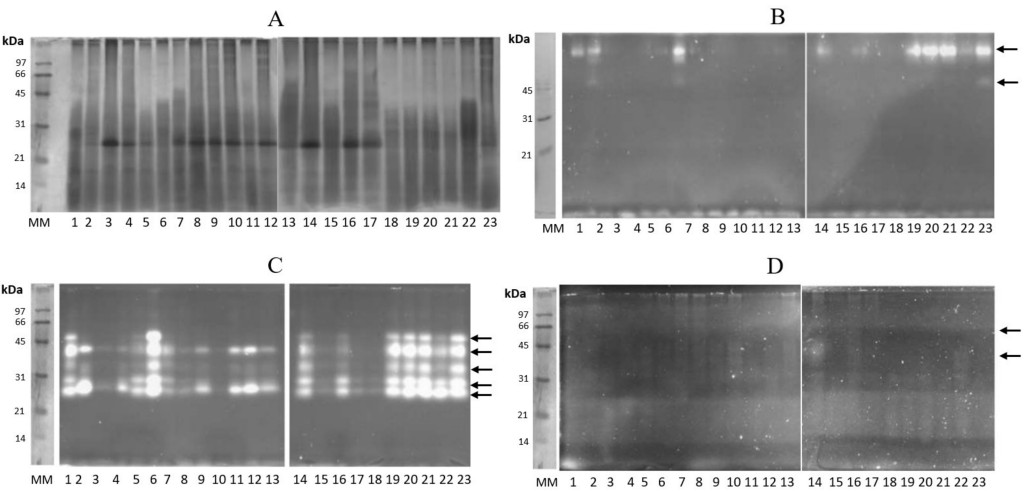

**Figure 4** **Protein profiling and zymography of *Stomolophus* sp. 2 gastric pouch contents.** (A) Protein profile in the gastric pouches of each of the 23 organisms sampled, (B) zymogram of $\beta$-$N$-acetylhexosaminidase activity, (C) zymogram of $\beta$-glucosidase activity, and (D) chitinase activity zymogram. 1–23 represent samples from gastric pouches from *Stomolophus* sp. 2 organisms. MM, Molecular weight marker, and white bands indicate hydrolysis of the substrate. The black arrows indicate the position of active enzymes.

**Table 4** **Inter-locality variation of *Stomolophus* sp. 2 gastric pouches glycosidase activity.**

| Zone | Enzymatic activity (mU/g of tissue) | | |
|---|---|---|---|
| | **Chitinase** | **$\beta$-$N$-acetiyhexosaminidase** | **$\beta$-glucosidase** |
| Las Guásimas | $36.89 \pm 0.90$[a] | $3848.24 \pm 156.56$[b] | $2195.44 \pm 84.21$[c] |
| Hermosillo | $25.15 \pm 0.39$[b] | $2988.57 \pm 87.96$[a] | $871.00 \pm 22.80$[a] |
| Caborca | $21.82 \pm 0.21$[c] | $3261.73 \pm 281.11$[a] | $1163.63 \pm 26.72$[b] |

**Notes.**
Different letters in each column indicate statistically different groups.

**Table 5** **Inter-locality variation factors of glycosidase activity.**

| Zone | Diameter of bell[*] (cm) | Superficial water temperatura (°C) | Chlorophyll $\alpha$ (mg/m$^3$) |
|---|---|---|---|
| Las Guásimas | $12.2 \pm 0.8$[c] | 20.6 | 3.4 |
| Hermosillo | $7.9 \pm 1.1$[a] | 23.4 | 2.6 |
| Caborca | $9.1 \pm 0.8$[b] | 24.2 | 3.4 |

**Notes.**
*Different letters in the column indicate statistically different groups. Chlorophyll $\alpha$ data was obtained from GIOVANNI 4.21.

was lower in Las Guásimas (20.6 °C) than in Hermosillo and Caborca (23.4 and 24.2 °C, respectively) (Table 5).

Evaluation of intra-local variation in three different geographic locations in Las Guásimas (Table 6) was to establish the difference between the enzymatic activity in the gastric pouches from jellyfish within a few kilometers of difference. The average bell diameter in the three sites and the temperature sea surface does not present a significant difference; the

**Table 6  Intra-locality variation of *Stomolophus* sp. 2 gastric pouch glycosidase activity.**

| Point of catch | Enzymatic activity (mU/g of tissue) | | |
|---|---|---|---|
| | Chitinase | $\beta$-$N$-acetylhexosaminidase | $\beta$-glucosidase |
| 1 | $20.11 \pm 0.13^a$ | $3054.54 \pm 127.53^d$ | $966.29 \pm 37.44^g$ |
| 2 | $23.82 \pm 0.30^b$ | $3385.59 \pm 118.63^e$ | $1254.27 \pm 28.52^h$ |
| 3 | $21.61 \pm 0.15^c$ | $2113.42 \pm 128.87^f$ | $702.55 \pm 24.32^i$ |

Notes.
Different letters indicate statistically different groups per column.

**Table 7  Intra-locality variation factors of glycosidase activity in gastric pouches from *Stomolophus* sp. 2.**

| Locality | Diameter of bell (cm) | Superficial water temperatura (°C) | Chlorophyll $\alpha$ (mg/m$^3$) |
|---|---|---|---|
| 1 | $10.5 \pm 1.1^a$ | 23.6 | 1.4 |
| 2 | $9.9 \pm 0.9^a$ | 23.6 | 1.3 |
| 3 | $93.0 \pm 0.8^a$ | 23.6 | 1.2 |

Notes.
Localities: 1, 27°46′17.46″N, 110°36′59.4″W; 2, 27°45′37.32″N, 110°36′45.84″W; 3, 27°43′59.4″N, 110°36′44.46″W. Chlorophyll $\alpha$ data was obtained from GIOVANNI 4.21.

concentration of chlorophyll $\alpha$ presents very similar values in the three points (Table 7). Data showed statistical differences between locations and glycolytic activity ($p < 0.05$); in intra-localities (1: 27°46′17.46″N, 110°36′59.4″O, 2: 27°45′37.32″N, 110°36′45.84″O) the chlorophyll $\alpha$ (1.4 and 1.3 mg/m$^3$, respectively) was slightly higher than in point 3 (27°43′59.4″N, 110°36′44.46″O), (1.2 mg/m$^3$).

# DISCUSSION

Jellyfish are considered carnivorous organisms. The specific digestion rate is related to the prey and predator size, the number of prey, differences in metabolic rates, digestive enzymes, and temperature (*Martinussen & Båmstedt, 1999*). In *Stomolophus* sp. 2, the digestion time differs in hours; the prey of smaller size disappear from the digestive system after 2 h of digestion, while some bivalves persist for up to 6 h (*Larson, 1991*). In jellyfish gastric pouches, the protein variability may be due to previous feeding or digestive stage; likewise, sex and age. Carbohydrate digestion in cannonball jellyfish is little known. The presence of chitinase (24.28 ±0.38 mU/g of dry tissue) and $\beta$-$N$-acetylhexosaminidase (3,645.00 ±198.10 mU/g of dry tissue) activity is related to the chitin content of the main species in the *Stomolophus* diet (shells of bivalves, gastropods, and crustaceans) since this carbohydrate represents 4.6% in dry weight of copepods and between 0.1 and 40% in mollusks (*Muzzarelli, Jeuniaux & Gooday, 1986*). The complete digestion of a polysaccharide like chitin requires at least one endo- and one exoglycosidase (*Matsumiya, Miyauchi & Mochizuki, 1998*). Both activities in the gastric pouches from *Stomolophus* sp. 2 indicate that the species is capable of digesting the chitin from the prey it ingests; this is important since the digestion rate in the *Stomolopus* genus depends on the prey type and can vary between 2 and 6 h (*Larson, 1991*).

By identifying $\beta$-glucosidase activity but not detecting cellulase activity (endoglucanase) and exoglucanase, it cannot be stated that the gastric pouch of *Stomolophus* sp. 2 can hydrolyze cellulose. However, $\beta$-glucosidase has also been reported to hydrolyze chitobiose molecules (disaccharides produced by the hydrolysis of chitin) and bonds of *N*-acetylmuramic acid (from the cell wall of bacteria) (*Hylleberg Kristensen, 1972*), suggesting that the $\beta$-glucosidase activity in jellyfish is associated with these biological functions.

Pectinase, carragenanase, xylanase, exo- and endo-glucanase, and amylase were not detected, coinciding with adult food preferences in *Stomolophus*. The amount of zooplankton is not significant since it represents around 1% of the ingested food (*Arai, 1997*). However, in polyps and early stages (Ephyra 1, Ephyra 5, Ephyra 10, Ephyra 15, and juvenile), amylase activity decreases significantly throughout its physiological development (*González-Valdovinos, Ocampo & Tovar-Ramírez, 2019*); this may explain the absence of amylase activity in the gastric pouches of adult *Stomolophus* sp. 2.

On the other hand, the absence of amylase activity contradicts the results obtained by *Bodansky & Rose (1922)*, where they found amylase and maltase activity in samples of *S. meleagris*. This difference may also be due to the incubation period in the enzymatic extract from *Stomolophus* sp. 2 gastric pouches (48 h *versus* 24 h in this study). Another aspect to consider is that the digestive enzyme activity in cnidarians such as *A. aurita* and sea anemones (*Adamsia palliata, Anemonia sulcata, Anthopleura balli*) only occurred in contact between food and gastrodermis and not in gastric fluids (*Bumann & Kuzirian, 1996*; *Jeuniaux, 1962*).

The individual analysis of every 23 organisms from various sample sites shows differences between enzymatic activity (Fig. 2). This phenomenon was similar in *A. aurita* due to organism variability in the digestion rate under controlled conditions (*Båmstedt & Martinussen, 2000*). Another critical factor is the last food intake since the digestive enzymatic activity in Cnidarians is higher after feeding (*Muscatine & Lenhoff, 1974*). In other marine species, such as lobster *Jasus edwardsii*, food requirements vary according to the ontogenic stage, causing carbohydrases (amylase, $\alpha$-glucosidase, cellulase, and chitinase) to be higher in lobsters (*Johnston, 2003*). In contrast with cnidarians and *Stomolophus* sp 2, the glycolytic activity is low compared to other marine invertebrates, such as annelids, crustaceans, mollusks (bivalves, gastropods, and echinoderms) and tunicates (*Molodtsov & Vafina, 1972*).

In the coastal lagoon Las Guasimas (Sonora, Mexico), the salinity varies from 35 to 41 ppt (*Padilla-Serrato et al., 2013*). Jellyfish are osmoconformers organisms, meaning that the internal ionic strength changes depending on the concentration of salts in the seawater in which they are found. Glycosidases from *Stomolophus* sp. 2 can be affected by ionic strength changes. The increased glycolytic activity in concentrations up to 1.5 M NaCl in *Stomolophus* sp. 2 suggests halotolerant enzymes (show activity of 0–4 M NaCl), similar to the digestive lipase/esterase activity in cannonball jellyfish reported by *Martínez-Pérez et al. (2020)*.

As well as ionic strength, other factors, such as pH, can modify the digestive glycolytic hydrolase activity. In some marine herbivorous organisms and fishes, digestion occurs at a

slightly acidic pH of 6.8 (*Solovyev et al., 2015*). In *Stomolphus* sp. 2, the gastric pouch average pH is 6.79, which indicates that the highest glycolytic activity was obtained at pH values lower than the natural gastric pouch pH. The three enzyme activities detected (chitinase, $\beta$-glucosidase, and ß-$N$-acetylhexosaminidase) in *Stomolophus* sp. 2 were higher at pH 5.0, and similar results have been reported in *Rhopilema asamushi*, *Palythoa caribaeroum*, and *Homarus americanus* for $\beta$-$N$-acetylhexosaminidase; *H. americanus* for $\beta$-glucosidase, and *Penaeus monodon* and *Artemia* spp. for chitinases (*Brockerhoff, Hoyle & Hwang, 1970*; *Funke & Spindler, 1989*; *Nagai, Watarai & Suzuki, 2001*; *Proespraiwong, Tassanakajon & Rimphanitchayakit, 2010*; *Souza et al., 2008*). It also coincides with what was reported in mollusks, where the optimum pH is neutral or slightly acidic (*Trincone, 2013*).

The optimum temperatures were 30 °C for chitinase activity, 40 °C for $\beta$-$N$-acetylhexosaminidase activity, and 50 °C for $\beta$-glucosidase from *Stomolophus* sp. 2 (Fig. 3); these temperature values are higher than the mean surface temperature (23 °C) of the jellyfish habitat. The enzymatic activities express at least 50% residual activity under environmental conditions. The $\beta$-$N$-acetylhexosaminidase and $\beta$-glucosidase activities showed lower activity at low temperatures. They maintained over 23% of their activity at 80 °C, which is interesting since most proteins are denatured at high temperatures, only a few degrees above the organism's ambient temperature. The temperature where the catalysis of $\beta$-$N$-acetylhexosaminidase is higher (40 °C), coincides with other marine species, such as the coral *Palythoa caribaeroum* and the marine mollusk *Patinopecten yessoensis* (*Sakai, Nakanishi & Kato, 1993*; *Souza et al., 2008*). In Artemia, the optimum temperature for chitinase activity was 55 °C (*Funke & Spindler, 1989*), higher than that identified in *Stomolophus* sp. 2.

Chitinase activity decreased at temperatures above 50 °C. This characteristic is shared with the glycosidases from some bivalves, which have an optimum temperature between 24 °C and 32 °C (*Brock, Kennedy & Brock, 1986*). On the other hand, the $\beta$-$N$-acetylhexosaminidase and chitinase activity did not differ between years or areas of the intra-locality, but different activities were observed in inter-localities.

The annual difference in the average size of the organisms can be ruled out as a direct factor on this enzymatic activity, but not the chlorophyll $\alpha$ concentration and surface temperature, which turned out to be different in both years, since these differences are reflected in a change in the enzymatic activity. Some authors have suggested a relationship between algal blooms and medusas proliferation. Studies suggest a trend towards a continued increase in the abundance of microalgae, as well as hypoxic conditions, could favor jellyfish growth. Indeed, nutrient enrichment in coastal waters may increase the biomass of phytoplankton, which could favor a greater abundance of zooplankton, which is the primary food source of jellyfish (*Giussani et al., 2016*). Therefore, the higher abundance of phyto- and zooplankton may affect the enzymatic activity of jellyfish glycosyl hydrolases, as shown in Tables 2 and 3. In the cnidarian anemone *Entacmaea medusivora* (Actiniaria, Anthozoa), which feeds on the golden jellyfish (*Mastigias papua*; Rhizostomeae, Scyphozoa), harboring an endosymbiotic dinoflagellate of the genus *Cladocopium* (Symbiodiniaceae), it has been shown that the digestive enzymes of the anemone cannot digest this type of microalgae. However, in juvenile jellyfish of the

*Phyllorhiza punctata,* a diet rich in rotifer and microalgae resulted in greater survival and growth than organisms fed with rotifers (*Miranda et al., 2016*).

The expression of $\beta$-$N$-acetylhexosaminidase activity is attributed to feeding by predation of various species (bivalves, copepods, gastropods, and barnacles) (*Álvarez Tello, López-Martínez & Lluch-Cota, 2016*; *Larson, 1991*; *Padilla-Serrato et al., 2013*), then it is assumed that there are no significant changes in the intake of this carbohydrate in the same geographical location, even in a different year of capture. The differences in digestive enzyme activity from gastric pouches from *Stomolophus* sp. 2 are due to different localities in the Gulf of California, which change from seasonal surface warming and are impacted by upwellings of different seasons (*Castro & Durazo, 2010*). These upwellings substantially affect the levels of diversity and abundance of many species of the pelagic ecosystem (*Castro & Durazo, 2010*), which cause variation in the food available for *Stomolophus* sp. 2.

Variations that respond to biological and environmental factors are influenced by the diversity of zooplanktonic species used as food, which can be observed with reports from different years and geographic locations (*Álvarez Tello, López-Martínez & Lluch-Cota, 2016*; *Padilla-Serrato et al., 2013*).

The $\beta$-glucosidase activity in different locations cannot be attributed to available prey but to the intake of phytoplankton and organic matter (*Arai, 1997*; *Nosrati et al., 2013*). The organic matter comes from marine microbial organisms, transport of organic matter carried by rainwater to the coasts, atmosphere, or groundwater (*Hansell & Carlson, 2002*), or upwellings originated by natural and human activities from the Sonoran coast. These activities produce waters rich in nutrients towards the surface (*Castro & Durazo, 2010*), changing the polysaccharides concentration in the surface seawater between 0.3 and 210 $\mu$M of carbon (*Gui-Peng et al., 2010*). This phenomenon is reflected in the annual variation of the chlorophyll $\alpha$ concentration since this difference is not statistically significant compared to the year with the highest concentration, where ß-glucosidase activity is more significant (Tables 2–3).

## CONCLUSION

The cannonball jellyfish (*Stomolophus* sp. 2) can synthesize digestive glycolytic hydrolases (chitinase, $\beta$-Glucosidase, and $\beta$-$N$-acetylhexosaminidase) in their gastric pouches, which are halotolerant enzymes with activity from 0–4 M NaCl and can work at optimum pH 5.0, and temperatures between 30–50 °C. The activity of digestive glycosidases varies due to jellyfish geographical localization in the Gulf of California, between different inter-localities on the Sonoran coast (Hermosillo, Las Guásimas, and Caborca). Due to the results obtained, the geographical location was the only factor of statistically significant differences in the glycolytic activity from the gastric pouch of cannonball jellyfish (*Stomolophus* sp.2), unlike the year of capture nor intra-local areas, which did not influence on the enzymatic activity found in the jellyfish. Also, no differences were identified between the years (2015–2016) and inter-localities.

## ACKNOWLEDGEMENTS

We are grateful to Rodolfo Navarro and Everardo Miranda from the Regional Center for Aquaculture and Fisheries Research, Instituto Nacional de Pesca y Acuacultura, for their help during sample collection.

### Funding

This work was supported by the Instituto Tecnológico de Sonora through Programa de Fomento y Apoyo a Proyectos de Investigación (PROFAPI). There was no additional external funding received for this study. The funders had no role in study design, data collection and analysis, decision to publish, or preparation of the manuscript.

### Grant Disclosures

The following grant information was disclosed by the authors:
Instituto Tecnológico de Sonora through Programa de Fomento y Apoyo a Proyectos de Investigación (PROFAPI).

### Competing Interests

The authors declare there are no competing interests.

### Author Contributions

- Raul Balam Martinez-Perez conceived and designed the experiments, performed the experiments, analyzed the data, prepared figures and/or tables, authored or reviewed drafts of the article, and approved the final draft.
- Jorge A. Rodriguez conceived and designed the experiments, performed the experiments, authored or reviewed drafts of the article, and approved the final draft.
- Miguel A. Cisneros-Mata conceived and designed the experiments, prepared figures and/or tables, and approved the final draft.
- Luis Alonso Leyva Soto performed the experiments, prepared figures and/or tables, and approved the final draft.
- Pablo Gortáres-Moroyoqui analyzed the data, prepared figures and/or tables, and approved the final draft.
- Ana Renteria-Mexia analyzed the data, prepared figures and/or tables, and approved the final draft.
- Edna Abigail Hernandez Corral conceived and designed the experiments, performed the experiments, authored or reviewed drafts of the article, and approved the final draft.
- Lourdes M. Diaz-Tenorio conceived and designed the experiments, performed the experiments, authored or reviewed drafts of the article, and approved the final draft.

### Data Availability

The raw data are available in Supplemental Files.

## Supplemental Information

Supplemental information for this article can be found online at http://dx.doi.org/10.7717/peerj.16417#supplemental-information.

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
