# Peer review of "Digestive glycosidases from cannonball jellyfish (Stomolophus sp. 2): identification and temporal-spatial variability"

_PeerJ, doi:10.7717/peerj.16417_

## Round 0.1 · original submission · Major Revisions

Please address the critiques of all reviewers and amend the manuscript accordingly.

Reviewer 1 ·

Basic reporting

English is clear, although I suggest that the professional staff evaluates if further copywriting is required. Adequate references, there is not much in the literature specific to this species.

Experimental design

It is unclear for each determination if the 23 sample tissues were pooled and used for one determination or if experimental replicates were made. The actual sampling method must be described and detailed where the mean and SD numbers come from. The methods used are appropriate and described in sufficient detail to be reproduced. In the SDS-PAGE section, please describe how the “protein mix” was prepared.
In line 168 it may refer to a Gel Doc EZ system more than a kit since the gel was manually stained in the old fashion way.
Line 184, please describe the preparation of the Lugol stain.
In lines 186-190, please describe how the GIOVANNI and MODIS services are accessed via a website. Or they are software that needs to be requested or licensed? More details, please.
In the statistical analysis section, much more detail about sampling and replicates is needed. The statement in lines 200-201 could be rewritten, such as “The protein content of the gastric pouches was 40.73 ¬± 2.47 mg” since the concept of “protein variability” is misleading.
The inclusion of table 1 is gratuitous since all the values are stated in lines 210-216, so either the text or the table should be included.
The interpretation of figure 4 is complicated; the identification of patterns in the zymogram is dubious due to the low abundance of the enzyme, or maybe more sample is needed. Also, the pattern looks identical to me in all lanes (4D).
The claim in line 330 that chitinases are denatured at temperatures higher than 50 C seems unsustain. The experiment is not looking for folded protein, just for the temperature dependence of enzymatic activity.
Please support the causality between chlorophyll content and enzymatic activity claimed in lines 335-338.

Validity of the findings

A list of correlations between environmental factors and enzymatic activities needs to be taken into a more in-depth discussion. What is known in other systems?

Reviewer 2 ·

Basic reporting

Martinez-Perez et al. identified enzyme activities of chitinase, β-glucosidase, and β-N-acetylhexosaminidase from the gastric pouches of the cannonball jellyfish Stomolophus sp.2, which were obtained from different locations. These 3 glycosidase activities under different salt concentration, pH, and temperature conditions were further measured and the optimums were found out. The authors evaluated the temporal and spatial regulation of the enzyme activities and claimed that the geographical location was the only factor of statistically significant influence on glycolytic activity from the gastric pouch of cannonball jellyfish (Stomolophus sp.2).

The English language should be improved to ensure that an international audience can clearly understand the authors’ text. Some examples where the language could be improved include lines 31-34, 63-67, 86-92, 153-156, 160-162, 189-190, 218-220, 228-229, 235-238, 323-325, and 330 – the current phrasing makes comprehension difficult. Other language issues: line 45 and 47, "whit" should be "with"; line 80, "are" should be "were"; line 119, "mixed" should be "resuspended". I suggest the authors to have a colleague who is proficient in English and familiar with the subject matter review the manuscript, or contact a professional editing service.

Acknowledgments part is missing in this manuscript, the authors should include this part.

Experimental design

The most important issue in this manuscript is the data interpretation. For example, in Table 2, it was listed that the β-glucosidases activity is 374 mU/g of tissue in 2015 while in 2016 it is 2195 mU/g of tissue. However, the authors stated that they show no significant difference in line 235-236; in Table 6, it is clear that β-N-acetylhexosaminidase and β-glucosidase enzymatic acticities from point of catch 2 and 3 are significantly different, the authors should address this other than claiming “data showed no statistical difference …” in line 253.

A minor issue is in the method description, the authors should give sufficient details to ensure the reproducibility. Some examples in this regard include line 134-135; 176-178; and 182.

Validity of the findings

Thank you for providing the raw data, however, for the Electrophoresis Gel images, the authors should annotate each lane: the gel image in the body text and the raw data cannot be aligned easily; for the excel form, the author should use unified format for all the text and use English uniformly other than using Spanish somewhere for example “P valor” and “tejido”.

As I have noted above for the data interpretation issue, the conclusion may be modified.

Additional comments

I commend the authors for their extensive data set and years of detailed work. If there is a weakness, it is in the data interpretation as I have noted above which should be improved upon before Acceptance.

Reviewer 3 ·

Basic reporting

-The manuscript must be revised by a fluent English speaker before publication, also it requires extensive edition, there is a lot of grammar issue, incomplete sentences, etc.
Abstract do not reflect the variability of carbohydrase between different localities, authors should include it as well as a conclusion.

Specific comments
Line 28-29: Requires to improved witting of the MS. Suggestion: …a wide variety of prey (crustaceans, mollusk, bivalves, etc.) as well as dissolved carbohydrates present in marine waters.
Line 29. This study was focused to detect and quantify the activity of digestive glycosydases…
Line 30 Delete and its evaluation.
Line 38. Please modify as follow: At least five and two protein bands with activity
Line 39: change zymography by zymogram.
Line 45 indicate the type of chlorophyll measured. Alpha, etc.
Line 46. Missing units.
Line 48-50: Conclusions are referred to the capability of the enzymes to hydrolyze substrates, there is not conclusion about the variation in the localities.
Line 49: acid, please be precise how much acid based on the authors results.
Line 50: water temperature, please be precise what temperatures were evaluated.
Line 67: Authors should state the importance to study digestion in this specie, rather to indicate the lack of information.
Line 73. Same as above, authors indicate that because cnidarians are carnivores and limited information exist about glycosyl hydrolases studies should be performed, we encourage authors to explain the importance to know and understand digestion of commercial species, such as jellyfish.

Line 95: Authors indicate that they evaluate the temporal-spatial regulation of digestive enzymes, however, there is not evidence of this.

Paragraph 102-111: It will be good to include a map with the location, so it is possible to understand potential spatial changes related to temperature or other important parameters such as chlorophyll.
Line: 122. BSA, please include the full name, the catalog number and the company of the standard.
Line 127: it should say: endoglucanase activity…
Line 134-135: Please explain or describe better the agitation by time intervals, aliquots of the reaction were collected at those times?
Line 134: Did the authors measure absorbance or how did they measure activity? please specify and include wavelength and data of the equipment.
Line 139: pH 6.0
Line 142: Please include the equipment used for measurement of the activity, as well if it was microplate or if the authors used cuvettes, how much of the reaction was used for reading absorbance, also did they performed an standard curve of p-nitrophenol?
Line 148: Please include the information regarding to the volume used for measurement of the activity, as well as the equipment used.
Section of the effect of temperature
This section lacks important information:
-The effect of temperature was incubating the enzyme on those temperatures and then in regular temperature for enzyme activity or how did they perform this experiment? Same information should be included in the case of pH and ionic strength.
Correct as follow: 5.0- 10.0; pH 3.0-5.0; pH 7.0-9.0; pH 10.0

Line 161: Inconclusive sentence: A gel of 12% sodium dodecyl sulfate-polyacrylamide (SDS-PAGE).
Line 162: loading buffer.
Line 163: please indicate the catalogue number of the LMW marker.
Line 172. Delete second citation Bai et al 2013.
Line 173: missing pH of the buffer solution
Line 176: how 4-MUGLc was dissolved, please indicate in parenthesis
Line 178: modify as follow…. With constant agitation (100 rpm) for 10 min, then visualized with UV light (include wavelength) using a XXXX.
Paragraph 180-184: requires edition,
The amylase activity zymogram consisted of the separation of protein extracts on a 12% SDS-PAGE. Afterwards, the gel was washed with distilled water and transferred to a buffer solution of 50 mM sodium acetate pH 5.0 for 60 min, then the solution was discarded and replaced with a substrate solution (0.5% starch solution in 50 mM sodium acetate pH 5.0), incubated for 120 min at 37 C. Activity bands of amylase were detected after addition of Lugol stain (how much or concentration) (reference).
Paragraph 187-190: How the chlorophyll data were collected and in which season or locations? Please specify.
The results section contains discussion, authors should modify accordingly in order to maintain only results, since authors have a discussion section.
Line 202 to 204: Discussion
Line 207: change measured by detected.
Line 208-209: discussion
Line 211: 56.90 – 154.51 mU/g
Line 213: 6.85-41.82 mU/g
Line 1277.35-8865.63 mU/g
Line 211-216: Any of the data were significant different statistically? Please indicate in the text
Line 217-220. Include data and how many organisms present the behavior.
Line 223: what it means drastically?
Line 221-227: please include data if they were statistically different.
Line 237: ontogenetic stage of the collected organisms.
Line 237-238. The mean of the bell diameter of two Stomolophus sp. was statistically different (P< XXX) with 3.4 cm higher in 2016 than XXX.
Line 239-240: The difference in temperature were in all the places sampled, same issue for chlorophyll a.
Line 242: statistically different, please include the data, value and P-value.
Line 243: different sampling sites, which ones?
Line 244: no difference, include value and P-value.
Line 246, 253: a-chlorophyll
Line 254: not statistically difference, data and p-value required.
Discussion
Line 266: space in parenthesis required.
Line 268. Remove comma before Both and change by a dot.
Line 274: delete by what and the phrase can be change as follow:
Suggesting that the b-glucosidase activity in jellyfish is associated with these biological functions.
Line 285: incubation period of XXXX missing information.
Line 287: sea anemones include scientific name to be homogeneous in the text
Extensive English edition is required in the discussion section, the information contained is important, but the redaction requires edition.
Line 362- synthesize instead of can produce.
Figure legend table 1. ND not detected but, in the table, authors used N/D, please modify accordingly.
Table 2. Indicate statistical differences, probably on b-glucosidase.
Table 3. What about the variation by location? Do the authors have any information? It could be important for the discussion section. Also include statistically difference if present.
Legend Fig. 1 B. It should say Protein (mg/ g of tissue)
Figure 3c, include the y axis for all the graphs.
Figure 4. Please indicate the meaning of the arrows

Experimental design

There are a lot of missing information on methods that require to complete it. See the basic reporting.

Validity of the findings

The data are important for the field of fisheries and aquaculture of cannonball.

Additional comments

-The manuscript must be revised by a native English speaker before publication, also it requires extensive edition, there is a lot of grammar issue, incomplete sentences, etc.
Abstract do not reflect the variability of carbohydrase between different localities, authors should include it as well as a conclusion.

Specific comments
Line 28-29: Requires to improved witting of the MS. Suggestion: …a wide variety of prey (crustaceans, mollusk, bivalves, etc.) as well as dissolved carbohydrates present in marine waters.
Line 29. This study was focused to detect and quantify the activity of digestive glycosydases…
Line 30 Delete and its evaluation.
Line 38. Please modify as follow: At least five and two protein bands with activity
Line 39: change zymography by zymogram.
Line 45 indicate the type of chlorophyll measured. Alpha, etc.
Line 46. Missing units.
Line 48-50: Conclusions are referred to the capability of the enzymes to hydrolyze substrates, there is not conclusion about the variation in the localities.
Line 49: acid, please be precise how much acid based on the authors results.
Line 50: water temperature, please be precise what temperatures were evaluated.
Line 67: Authors should state the importance to study digestion in this specie, rather to indicate the lack of information.
Line 73. Same as above, authors indicate that because cnidarians are carnivores and limited information exist about glycosyl hydrolases studies should be performed, we encourage authors to explain the importance to know and understand digestion of commercial species, such as jellyfish.

Line 95: Authors indicate that they evaluate the temporal-spatial regulation of digestive enzymes, however, there is not evidence of this.

Paragraph 102-111: It will be good to include a map with the location, so it is possible to understand potential spatial changes related to temperature or other important parameters such as chlorophyll.
Line: 122. BSA, please include the full name, the catalog number and the company of the standard.
Line 127: it should say: endoglucanase activity…
Line 134-135: Please explain or describe better the agitation by time intervals, aliquots of the reaction were collected at those times?
Line 134: Did the authors measure absorbance or how did they measure activity? please specify and include wavelength and data of the equipment.
Line 139: pH 6.0
Line 142: Please include the equipment used for measurement of the activity, as well if it was microplate or if the authors used cuvettes, how much of the reaction was used for reading absorbance, also did they performed an standard curve of p-nitrophenol?
Line 148: Please include the information regarding to the volume used for measurement of the activity, as well as the equipment used.
Section of the effect of temperature
This section lacks important information:
-The effect of temperature was incubating the enzyme on those temperatures and then in regular temperature for enzyme activity or how did they perform this experiment? Same information should be included in the case of pH and ionic strength.
Correct as follow: 5.0- 10.0; pH 3.0-5.0; pH 7.0-9.0; pH 10.0

Line 161: Inconclusive sentence: A gel of 12% sodium dodecyl sulfate-polyacrylamide (SDS-PAGE).
Line 162: loading buffer.
Line 163: please indicate the catalogue number of the LMW marker.
Line 172. Delete second citation Bai et al 2013.
Line 173: missing pH of the buffer solution
Line 176: how 4-MUGLc was dissolved, please indicate in parenthesis
Line 178: modify as follow…. With constant agitation (100 rpm) for 10 min, then visualized with UV light (include wavelength) using a XXXX.
Paragraph 180-184: requires edition,
The amylase activity zymogram consisted of the separation of protein extracts on a 12% SDS-PAGE. Afterwards, the gel was washed with distilled water and transferred to a buffer solution of 50 mM sodium acetate pH 5.0 for 60 min, then the solution was discarded and replaced with a substrate solution (0.5% starch solution in 50 mM sodium acetate pH 5.0), incubated for 120 min at 37 C. Activity bands of amylase were detected after addition of Lugol stain (how much or concentration) (reference).
Paragraph 187-190: How the chlorophyll data were collected and in which season or locations? Please specify.
The results section contains discussion, authors should modify accordingly in order to maintain only results, since authors have a discussion section.
Line 202 to 204: Discussion
Line 207: change measured by detected.
Line 208-209: discussion
Line 211: 56.90 – 154.51 mU/g
Line 213: 6.85-41.82 mU/g
Line 1277.35-8865.63 mU/g
Line 211-216: Any of the data were significant different statistically? Please indicate in the text
Line 217-220. Include data and how many organisms present the behavior.
Line 223: what it means drastically?
Line 221-227: please include data if they were statistically different.
Line 237: ontogenetic stage of the collected organisms.
Line 237-238. The mean of the bell diameter of two Stomolophus sp. was statistically different (P< XXX) with 3.4 cm higher in 2016 than XXX.
Line 239-240: The difference in temperature were in all the places sampled, same issue for chlorophyll a.
Line 242: statistically different, please include the data, value and P-value.
Line 243: different sampling sites, which ones?
Line 244: no difference, include value and P-value.
Line 246, 253: a-chlorophyll
Line 254: not statistically difference, data and p-value required.
Discussion
Line 266: space in parenthesis required.
Line 268. Remove comma before Both and change by a dot.
Line 274: delete by what and the phrase can be change as follow:
Suggesting that the b-glucosidase activity in jellyfish is associated with these biological functions.
Line 285: incubation period of XXXX missing information.
Line 287: sea anemones include scientific name to be homogeneous in the text
Extensive English edition is required in the discussion section, the information contained is important, but the redaction requires edition.
Line 362- synthesize instead of can produce.
Figure legend table 1. ND not detected but, in the table, authors used N/D, please modify accordingly.
Table 2. Indicate statistical differences, probably on b-glucosidase.
Table 3. What about the variation by location? Do the authors have any information? It could be important for the discussion section. Also include statistically difference if present.
Legend Fig. 1 B. It should say Protein (mg/ g of tissue)
Figure 3c, include the y axis for all the graphs.
Figure 4. Please indicate the meaning of the arrows

---

## Round 0.2 · Minor Revisions

Please address remaining issues pointed by the reviewers and revise manuscript accordingly.

Reviewer 1 ·

Basic reporting

The Stomolophus jellyfish genus comprises five species (S. agaricus, S. chunii, S. collaris, S. fritillaria, and S. meleagris), and a more recently described Stomolophus sp. 2.


Please cite López-Martínez, J., Álvarez-Tello, F. J., Porchas-Cornejo, M. A., Nevárez-López, C. A., Muhlia-Almazán, A., & Urías-Padilla, K. V. (2023). Multiple reproduction forms in the polyps of the cannonball jellyfish Stomolophus sp. 2: Probable life-cycle reversal. Journal of Experimental Zoology Part A: Ecological and Integrative Physiology, 339, 239– 252. https://doi.org/10.1002/jez.2673 as the description of this novel species.
Or another suitable reference to this "sp.2" species-.

Experimental design

Can you provide the taxonomical keys to identify Stomolophus sp. 2 after capture?

Also, how was the standard deviation (SD) calculated for the enzymatic activities? It needs to be mentioned whether triplicates were considered, and this information is necessary for accuracy.

Validity of the findings

Can you provide more information on how the location impacts enzymatic activity during different seasons in 2015-2016? Is it possible that the availability of biota as a food source is a contributing factor leading to adaptation? Kindly provide more details on this.

Additional comments

If "Figure extra" is included in the document, it must be numbered. Alternatively, it can be declared as supplemental materials.

Reviewer 2 ·

Basic reporting

no comment

Experimental design

no comment

Validity of the findings

no comment

Additional comments

The authors have revised and modified the old manuscript largely as I have suggested. However, there are still some areas where the authors' attention is needed, including the ones listed below:
line 93, ingest should be ingests;
line 172, the author may explain what's the meaning of "...buffer for the divide...";
the sentence in line 244-246 is still hard to understand.
line 363, please make clear what is "The optimal temperature activity of ..."
line 193, "a 100 mM citrate buffer, pH 5.0 with 2.5% of Triton X-100 for 60 min," should be changed to "a buffer containing 100 mM citrate, pH 5.0, 2.5% Triton X-100 for 60 min,"
line 206, "05%" should be "0.5%"

---

## Round 0.3 · accepted · Accept

Thank you for addressing the remaining issues and revising your manuscript in line with the recommendations of the reviewers. Your revised manuscript is acceptable now.